# The association of D-dimer with clinicopathological features of breast cancer and its usefulness in differential diagnosis: A systematic review and meta-analysis

**Yan Lu, LongYi Zhang**[ID]**\*, QiaoHong Zhang, YongJun Zhang, DeBao Chen, JianJie Lou, JinWen Jiang, ChaoXiang Ren**

Clinical Laboratory, Dongyang People's Hospital, Dongyang, Zhejiang, China

* happy_zhang1y@163.com

## Abstract

### Background

Studies have shown that D-dimer levels are significantly correlated with the differential diagnosis and clinicopathological features of breast cancer. However, the results are currently limited and controversial. Therefore, we performed this meta-analysis to evaluate the relationship between D-dimer levels and breast cancer.

### Materials and methods

The PubMed, Embase, Cochrane Library, China National Knowledge Infrastructure, Chinese Biomedical Literature, and Wanfang databases were searched to find studies that assessed the association of D-dimer with clinicopathological features of breast cancer and its usefulness in aiding with differential diagnosis. The standardized mean difference (SMD) was applied as the correlation measure.

### Results

A total of 1244 patients with breast cancer from 15 eligible studies were included in the meta-analysis. D-dimer levels were higher in the breast cancer group than in the benign (SMD = 1.02; 95% confidence interval [CI] = 0.53–1.52) and healthy (SMD = 1.27; 95% CI = 0.85–1.68) control groups. In addition, elevated D-dimer levels were associated with progesterone receptor-negative tumors (SMD = -0.25; 95% CI = -0.44–-0.05). Similarly, there was a significant correlation between D-dimer levels and tumor node metastasis staging (n = 11, SMD = 0.82; 95% CI = 0.57–1.06) and lymph node involvement (n = 8, SMD = 0.79; 95% CI = 0.50–1.09). In contrast, other clinicopathological factors, including estrogen receptor expression and human epidermal growth factor receptor 2 expression, were not associated with D-dimer levels.

**Data Availability Statement:** All relevant data are within the manuscript and its Supporting Information files.

**Funding:** The authors received no specific funding for this work.

**Competing interests:** The authors have declared that no competing interests exist.

## Conclusion

The results of this meta-analysis indicate that plasma D-dimer levels can be used as an important reference for the early identification and staging of breast cancer.

## Introduction

Breast cancer is the leading cause of death in women aged between 20 and 59 years and is estimated to account for 30% of all new cancer diagnoses in women in 2019[1]. Breast cancer has multiple levels of tumor heterogeneity. Clinical pathological conditions such as tumor node metastasis (TNM) stage, hormone receptor expression, human epidermal growth factor 2 (HER2) expression, and metastasis lead to different prognoses of breast cancer[2]. From 1990 to 2016, the mortality rate of female breast cancer decreased by 40% [1], but it still threatens women's health. Early diagnosis and treatment are key to improving the survival rates of breast cancer[3]. In addition to clinically and widely used tumor markers, such as carcinoembryonic antigen[4] and cancer antigen 15–3[5], other clinical laboratory indicators are urgently needed to assist in differential diagnosis and predict prognosis.

Tumor-induced coagulation is closely related to tumorigenesis and tumor development. Malignant disease can show signs of venous thromboembolism years before the patient has any obvious clinical symptoms[6]. By promoting neovascularization and metastasis, a vicious cycle is formed between procoagulant proteins and malignant tumor cells[7]. There is evidence that activated fibrinogens prevent NK cell-mediated tumor cell elimination, improve circulating tumor cell survival, increase tumor metastasis potential, and lead to poor prognosis[8]. Therefore, D-dimer, which is the end product of fibrinogen hydrolysis, has certain clinical value for the differential screening of benign and malignant tumors[9] and prediction of the prognosis of tumors[10–12]. Studies have shown that D-dimer has a significant correlation with the diagnosis and prognosis of a variety of malignant tumors (e.g., colorectal cancer and ovarian cancer), and D-dimer levels can be used as a diagnostic marker to design more individualized and effective treatment strategies [13].

However, Research on evaluating the association of D-dimer levels with breast cancer are currently limited, and the results have been controversial. Therefore, this meta-analysis was performed to assess the association between D-dimer levels and breast cancer-associated differential diagnosis and clinicopathological features.

## Materials and methods

### Literature search

The literature search was performed using the PubMed, Cochrane Library, Embase, China National Knowledge Infrastructure, Chinese Biomedical Literature, and Wanfang databases. We included articles published from the establishment of the database to March 19, 2019. We included only studies published in English or Chinese. The keywords used for the search can be found in S1 Table. We also performed a supplementary search for references included in the studies identified in the original search.

### Inclusion and exclusion criteria

The inclusion criteria were as follows: 1) the study group consisted of patients with breast cancer with a definite diagnosis; 2) the control group consisted of healthy women or patients with

benign breast tumors; 3) the D-dimer test method in the study was clear; 4) the study results contained or had sufficient data to calculate the mean and standard deviation, defined here as more than 20 patients; and 5) the study showed a correlation between D-dimer levels and diagnostic and/or clinicopathological features of breast cancer. The exclusion criteria were as follows: 1) case reports or reviews; 2) studies describing animal experiments; 3) repeated publications; and 4) articles with a low Newcastle-Ottawa scale (NOS) score (≤4).

### Data extraction and quality assessment

Data extraction and quality evaluation of the literature were performed independently by two authors. We extracted the following information: first author's last name, year of publication, country, method used to assess D-dimer levels, type of anticoagulant used, number of experimental groups included in the study, number of healthy controls and benign tumor controls, and number of patients with TNM stage I-II and III-IV disease. The NOS standard[14] was used as a research quality assessment standard. Studies with a score ≤ 4 were considered low quality. When there was a difference in opinion on a document, the two authors resolved the problem through mutual discussion and requested help from a third author if necessary.

### Statistical analysis

All data analyses were performed using the Review Manager software version 5.3 (Cochrane Collaboration, London, UK) and STATA software version 14.0 (Stata Corporation, College Station, TX, USA). The standardized mean difference (SMD) was used as a measure of the association between D-dimer levels and breast cancer, and the results are presented in the form of forest plots. Inter-study heterogeneity was assessed using the Q test and $I^2$ statistic. When $P > 0.10$ or $I^2 < 50\%$ indicated that there was no obvious heterogeneity[15], a fixed effects model was used; otherwise, a random effects model was used[16]. In addition, when the heterogeneity was significant, we performed subgroup analyses, followed by a sensitivity analysis. We used a funnel plot and an Egger test to assess publication bias[17]. $P < 0.05$ was considered statistically significant.

## Results

### Study search

Through our database search, we found 619 studies, of which 474 remained after duplicates were excluded. Based on the title and abstract, we excluded 434 articles that were not related to the research content and evaluated the remaining 40 articles in full. After full-text articles were assessed for eligibility, we included 15 studies that could be used for meta-analysis. A flow chart of the screening process is shown in Fig 1.

### Characteristics of eligible studies

Table 1 summarizes the basic information of the 15 eligible studies. The included studies were published between 2000 and 2018. D-dimer detection methods included enzyme-linked immunosorbent assay, immunoturbidimetry, and enzyme-linked immunofluorescence.

### Outcomes

We first compared the breast and benign control groups. The benign control groups from 8 studies were stratified using the D-dimer test. The total effect rate showed that the D-dimer level was higher in the breast cancer group (SMD = 1.02; 95% confidence interval [CI] = 0.53–1.52; $P < 0.0001$)(Fig 2A).

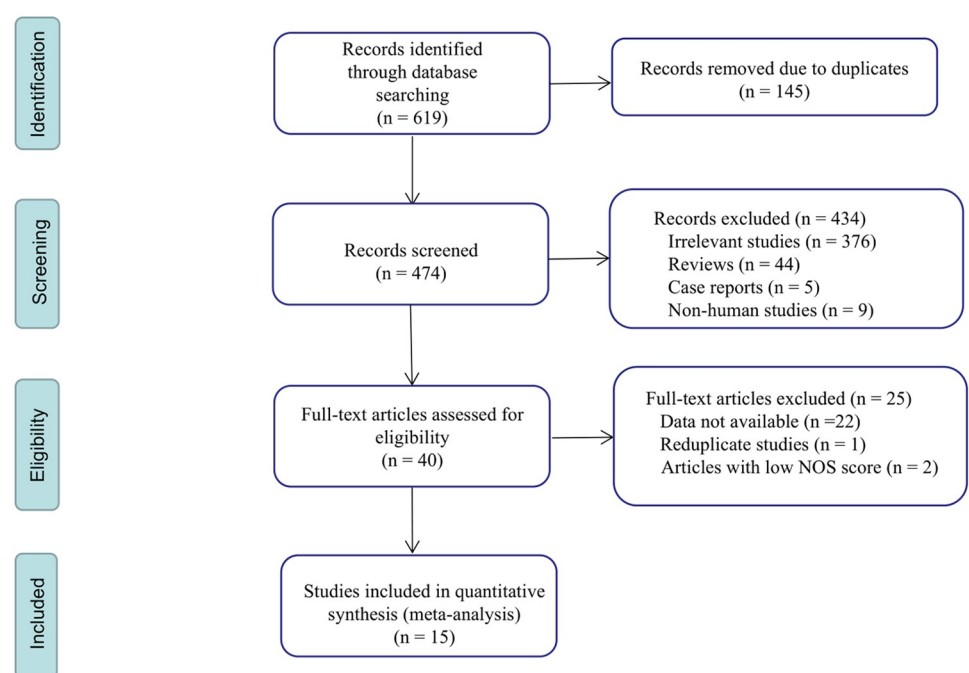

**Fig 1. Flow diagram of the study selection process.** NOS: Newcastle-Ottawa scale.

**Table 1. Characteristics of included studies.**

| Author | Year | Country | Detection Method/Anticoagulant | Breast cancer patients | Benign controls | Healthy controls | TNM stage I-II | TNM stage III-IV | NOS score |
|---|---|---|---|---|---|---|---|---|---|
| Blackwell[18] | 2000 | USA | ELISA/sodium citrate | 95 | NR | NR | 69 | 26 | 7 |
| Hua[19] | 2004 | China | ELISA/ethylenediamine tetra-acetic acid | 51 | 10 | 42 | 40 | 11 | 7 |
| Kim[20] | 2004 | Korea | ITM/sodium citrate | 93 | 27 | 29 | 77 | 10 | 8 |
| Khangarot [21] | 2010 | India | ELISA/NR | 50 | NR | NR | 20 | 30 | 6 |
| Zhao[22] | 2011 | China | ITM/NR | 43 | 43 | 43 | 32 | 11 | 7 |
| Xie[23] | 2011 | China | ITM/sodium citrate | 95 | 80 | NR | 58 | 37 | 7 |
| Huang[24] | 2012 | China | ITM/sodium citrate | 149 | 89 | 82 | 87 | 62 | 8 |
| Zhou[25] | 2012 | China | ELISA/NR | 48 | 40 | 40 | 36 | 12 | 7 |
| Liu[26] | 2013 | China | ITM/sodium citrate | 142 | NR | 150 | NR | NR | 7 |
| Chaari[27] | 2014 | France | ELFA/sodium citrate | 62 | NR | 30 | NR | NR | 6 |
| Yang[28] | 2014 | China | ITM/sodium citrate | 59 | NR | 50 | 29 | 31 | 7 |
| Feng[29] | 2014 | China | ELFA/NR | 189 | NR | NR | 95 | 94 | 7 |
| Chai[30] | 2015 | China | ITM/sodium citrate | 73 | 36 | 50 | NR | NR | 7 |
| Bai[31] | 2017 | China | ITM/sodium citrate | 35 | 37 | NR | NR | NR | 5 |
| S.H.[32] | 2018 | India | ITM/sodium citrate | 60 | NR | NR | 40 | 20 | 7 |

ITM: immunoturbidimetry; ELISA: enzyme-linked immunosorbentassay; ELFA: enzyme-linked immunofluorescence assay; NOS: Newcastle-Ottawa Scale; NR: not reported

(A)

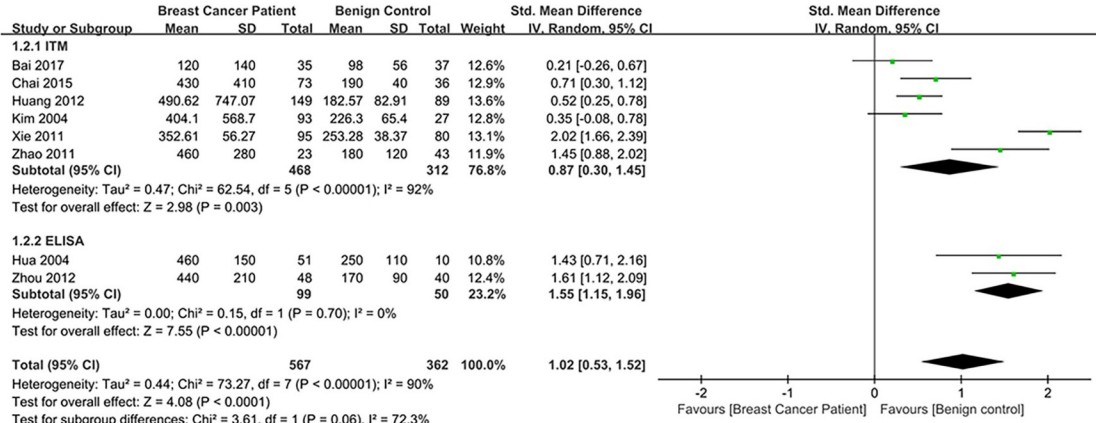

(B)

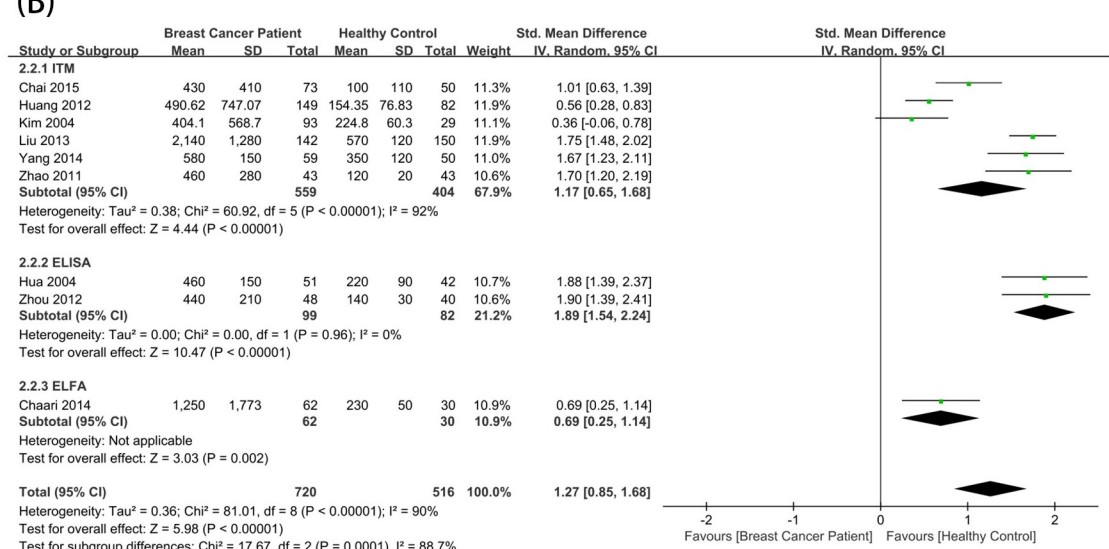

**Fig 2. Relationship between D-dimer levels and breast cancer diagnosis.** Forest plots depicting comparisons between breast cancer patients and (A) benign controls and (B) healthy controls. SD: standard deviation; CI: confidence interval; ELISA: enzyme-linked immunosorbent assay; ITM: immunoturbidimetry; ELFA: enzyme-linked immunofluorescence assay.

We next compared the breast and healthy control groups. After stratification using the D-dimer test, nine articles evaluating a healthy control group were divided into three subgroups. Using a random effects model, the total effect rate showed that the D-dimer level was significantly higher in the breast cancer group (SMD = 1.27; 95% CI = 0.85–1.68; $P < 0.00001$) (Fig 2B).

We also examined the correlation between D-dimer levels and clinical pathological parameters of breast cancer. Four studies examined the relationship between D-dimer levels and progesterone receptor (PR) expression, and there was no significant heterogeneity ($P = 0.38$, $I^2 = 3\%$) (Fig 3A). Using a fixed effects model, we observed that elevated D-dimer levels were associated with PR-negative tumors (SMD = -0.25; 95% CI = -0.44–-0.05; $P = 0.01$). There was also a significant correlation between D-dimer levels and TNM stage (n = 11, SMD = 0.82; 95% CI = 0.57–1.06; $P < 0.00001$) and lymph node involvement(n = 8, SMD = 0.79; 95% CI = 0.50–1.09, $P < 0.00001$) (Fig 3B and 3C). Here, we used a random effects model and

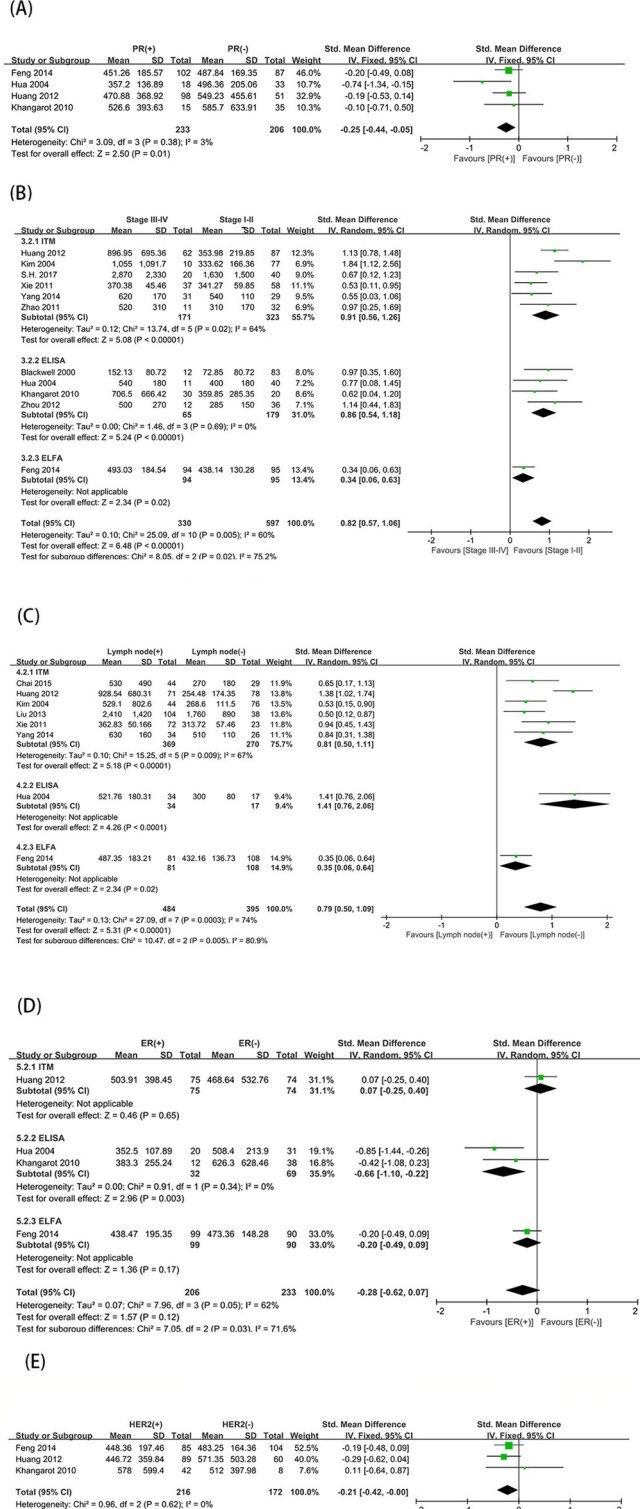

**Fig 3. Relationship between D-dimer levels and clinicopathological characteristics of breast cancer.** Forest plots of SMDs for the association between D-dimer and (A) progesterone receptor (PR) status (positive vs. negative), (B) tumor node metastasis (TNM) stage (stage III-IV vs. stage I-II), (C) lymph node status (positive vs. negative),(D) estrogen receptor (ER) status (positive vs. negative), and (E) human epidermal growth factor receptor (HER2) status (positive vs. negative). SMD: standardized mean difference; SD: standard deviation; CI: confidence interval.

combined subgroup analyses due to significant heterogeneity (TNM stage: $P = 0.005$, $I^2 = 60\%$; lymph node involvement: $P = 0.0003$, $I^2 = 74\%$). In contrast, other clinicopathological factors were not associated with D-dimer levels, including estrogen receptor (ER) expression (n = 4, SMD = -0.28; 95% CI = -0.62–0.07; $P = 0.12$) and HER2 expression (n = 3, SMD = -0.21; 95% CI = -0.42–0.00; $P = 0.05$) (Fig 3D and 3E). Due to the heterogeneity, the correlation between D-dimer levels and ER ($P = 0.05$, $I^2 = 62\%$) was based on a random effects model, and the correlation between D-dimer levels and HER2 ($P = 0.62$, $I^2 = 0\%$) used a fixed effects model.

## Heterogeneity

As shown in Fig 3, the subgroup analysis based on the differences in D-dimer detection methods found significant differences between the subgroups (benign controls, $I^2 = 72.3$; healthy controls, $I^2 = 88.7\%$; TNM, $I^2 = 75.2\%$; lymph node status, $I^2 = 80.9\%$; ER, $I^2 = 71.6\%$).

Additionally, most of the literature was obtained from China, and the sample sizes were smaller in other countries. The subgroup analysis was also used to examine the source of heterogeneity based on region. In addition to the benign control group ($I^2 = 79.2\%$) (Fig 4A), the results showed that there were no significant differences between the subgroups of the other groups with significant heterogeneity. (Fig 4B, 4C, 4D and 4E).

## Publication bias and sensitivity analysis

The symmetry of the funnel plot and results of the Egger's test (benign controls, $P = 0.470$; healthy controls, $P = 0.545$; TNM, $P = 0.093$; lymph node status, $P = 0.204$; PR, $P = 0.495$; ER, $P = 0.272$; HER2, $P = 0.408$) indicated that there was no publication bias. Sensitivity analysis was used to test the effect of a single study on the results. No significant differences were found when we removed any of the studies included in the analysis, indicating that the conclusions were stable.

## Discussion

To the best of our knowledge, this is the first meta-analysis on the role of D-dimer in the differential diagnosis and clinicopathological characteristics of breast cancer. As early as 1991, Mitter[33] found that D-dimer levels were elevated in patients with breast cancer. With the deepening of research in recent years, more links between D-dimer and the clinical pathology of breast cancer have been proposed.

### The role of D-dimer in the differential diagnosis of breast cancer

The results showed that the D-dimer level in the breast cancer group was significantly higher than those in the benign and healthy control groups. Increased plasma D-dimer levels reflect increased activation of the coagulation system in patients with breast cancer, suggesting that the plasma D-dimer level could have an auxiliary value for the differential diagnosis of breast cancer. Studies have shown that the sensitivity and specificity of D-dimer is higher than that of the existing tumor markers cancer antigen 15–3 and carcinoembryonic antigen [34]. Unfortunately, most of the research data did not allow to calculate the sensitivity and specificity of the effect indicator of D-dimer level for the diagnosis of breast cancer.

### The relationship between D-dimer and clinical pathology of breast cancer

Despite advances in breast cancer treatment, patients with metastatic breast cancer have a poor prognosis, with a low median survival of at most 2 to 3 years [2]. Plasma D-dimer levels in patients with TNM stage III-IV disease were significantly different from those in patients

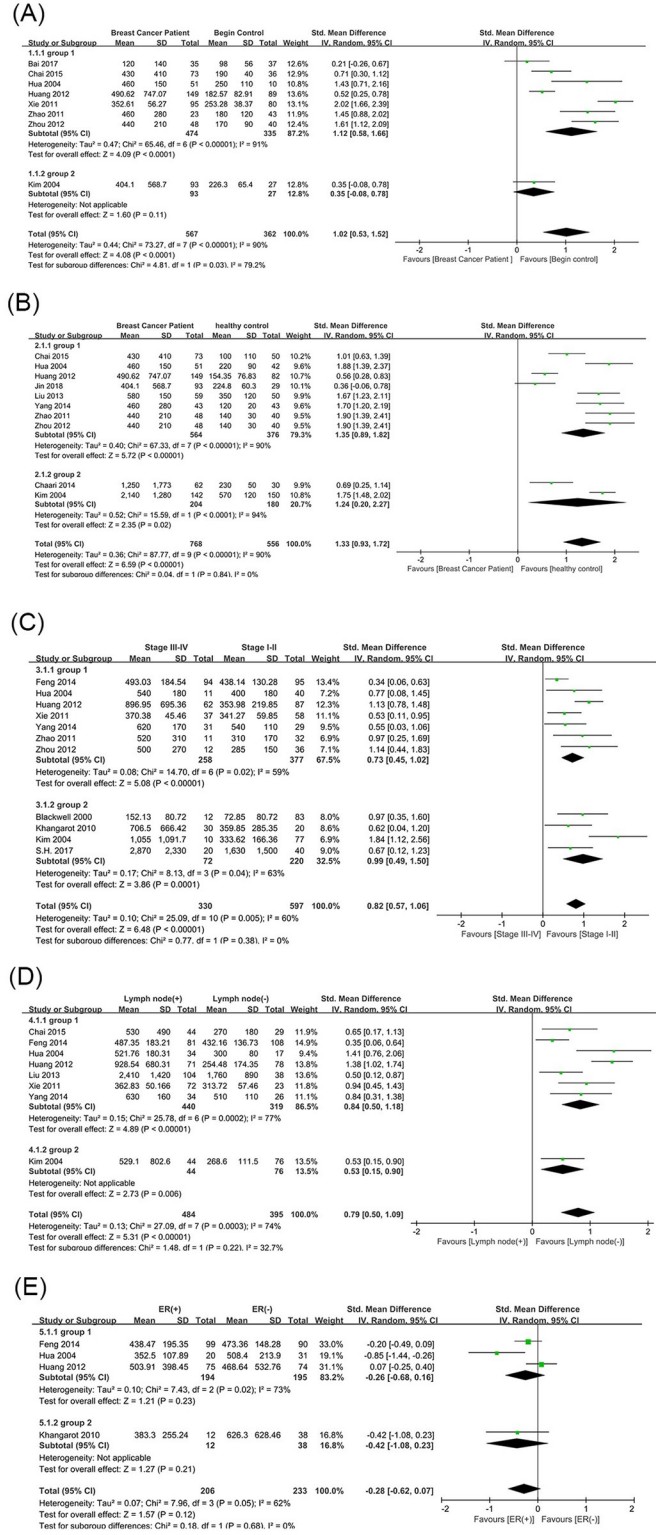

**Fig 4. Subgroup analysis of D-dimer levels and breast cancer-associated differential diagnosis and clinicopathological features according to region.** Plots depicting comparisons between breast cancer patients and (A) benign controls, (B) healthy controls, (C) tumor node metastasis (TNM) stage (stage III-IV vs. stage I-II), (D) lymph node status (positive vs. negative), and (E)estrogen receptor (ER) status (positive vs. negative). Group1: China; Group2: Regions outside China.

with stage I-II disease. Plasma D-dimer levels were also significantly higher in patients with lymph node metastasis than in patients without metastasis. Elevated D-dimer levels suggest a worsening of the disease, a later clinical stage, and a greater likelihood of tumor metastasis. The plasma D-dimer levels can be used as an auxiliary index for the diagnosis and staging of breast cancer. Furthermore, in this study, the D-dimer level was not related to the ER or HER2 status of patients with breast cancer, and it was increased in patients with PR-negative tumors. Due to the limitations of the literature, the role of D-dimer in the clinical pathology and prediction of prognosis of breast cancer still needs to be studied in a large number of patients.

## Limitations

The existence of heterogeneity is a potential problem when interpreting the results of this meta-analysis. To this end, we performed a subgroup analysis based on the differences in D-dimer detection methods. The results indicated that the difference in D-dimer detection methods is one of the main sources of heterogeneity. Because our meta-analysis is based on published research, the fact that most of the data coming from China may lead to regional bias. Therefore, the subgroup analysis was also used to examine the source of heterogeneity based on region with only significant differences in the benign control group. However, after excluding the study by Kim et al.[20] of Korea from the benign control group, the heterogeneity between the eight studies from China did not reduce, indicating that the regional differences cannot explain the heterogeneity between benign control groups. In addition, there may be other sources of heterogeneity. For example, this meta-analysis only included English and Chinese literature, which leads to language bias. Fortunately, although heterogeneity existed, the sensitivity analysis was stable, and no publication bias was found.

At present, there is no uniform standard for the methods and units used to detect D-dimer levels, and the consistency between the results of the same test items in each laboratory is not strong. In this paper, the unified D-dimer unit was ng/mL, and the standardized mean difference was used as the effect combination index. However, inconsistent detection methods, reagents and type of anticoagulant may cause the absolute D-dimer value to differ greatly, leading to high heterogeneity among the literature results. Therefore, a uniform methodological standard should be established for D-dimer detection so that the data between different laboratories can be interoperable or comparable.

## Conclusion

In this meta-analysis, plasma D-dimer levels were elevated in patients with breast cancer and correlated with PR expression, TNM stage, and metastasis in breast cancer. This evidence suggests that D-dimer has potential in the differential diagnosis and staging of breast cancer. However, the current results are somewhat restrictive, and we recommend further big data research and development of unified D-dimer detection methods in multiple regions.

## Supporting information

**S1 Table. Search strategy.**
(DOCX)

**S2 Table. Data of the present study.**
(XLSX)

**S1 Checklist. PRISMA checklist.**
(DOC)

## Author Contributions

**Conceptualization:** DeBao Chen, JinWen Jiang.

**Data curation:** Yan Lu, QiaoHong Zhang, YongJun Zhang.

**Methodology:** QiaoHong Zhang, JianJie Lou.

**Resources:** DeBao Chen, JianJie Lou.

**Software:** YongJun Zhang.

**Supervision:** ChaoXiang Ren.

**Validation:** JinWen Jiang, ChaoXiang Ren.

**Writing – original draft:** Yan Lu.

**Writing – review & editing:** LongYi Zhang.

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
