## [Decision Letter · Decision Letter 0]

18 Jun 2019

PONE-D-19-14829

The association of D-dimer with clinicopathological features of breast cancer and its usefulness in differential diagnosis: a systematic review and meta-analysis

PLOS ONE

Dear  Dr. Zhang,

Thank you for submitting your manuscript to PLOS ONE. After careful consideration, we feel that it is well organized and clear, and supply sufficient information regarding the topic focusing on breast cancer. However, the manuscript does not fully meet PLOS ONE’s publication criteria as it currently stands. Therefore, we invite you to submit a revised version of the manuscript that addresses the points raised during the review process.

We would appreciate receiving your revised manuscript by 29 July. To enhance the reproducibility of your results, we recommend that if applicable you deposit your laboratory protocols in protocols.io, where a protocol can be assigned its own identifier (DOI) such that it can be cited independently in the future. For instructions see: http://journals.plos.org/plosone/s/submission-guidelines#loc-laboratory-protocols

We look forward to receiving your revised manuscript.

Kind regards,

Elda Tagliabue

Academic Editor

PLOS ONE

Journal Requirements:

Reviewers' comments:

Reviewer's Responses to Questions

**Comments to the Author**

1. Is the manuscript technically sound, and do the data support the conclusions?

Reviewer #1: Partly

Reviewer #2: Partly

2. Has the statistical analysis been performed appropriately and rigorously? 

Reviewer #1: No

Reviewer #2: Yes

3. Have the authors made all data underlying the findings in their manuscript fully available?

Reviewer #1: Yes

Reviewer #2: Yes

4. Is the manuscript presented in an intelligible fashion and written in standard English?

Reviewer #1: Yes

Reviewer #2: Yes

5. Review Comments to the Author

Reviewer #1: Title: The association of D-dimer with clinicopathological features of breast cancer and its usefulness in differential diagnosis: a systematic review and meta-analysis (Zhang et al.)

The present manuscript is aimed at demonstrating the association between D-dimer and clinicopathological features (i.e. progesterone receptor-negative tumors, tumor node metastasis staging and lymph node involvement), by means of a meta-analysis based on 15 studies including 1244 patients with breast cancer.

MAJOR ISSUES

1. Figure 3E is identical to Figure 3A. Please correct the duplicated figure;

2. As stated from the authors in the Discussion, the literature of this meta analysis is limited and a large number of patients is still needed. Moreover, they are also aware that there is the risk of a regional bias, since almost all studies are from China. Results are “somewhat restrictive”, as stated in the Conclusion.

MINOR ISSUES

1. In the inclusion and exclusion criteria section, “the study results had a normal distribution” does not mean anything. Do the authors mean the distribution of the response variable?

Again, “the results were skewed” has no sense. Please explain it better;

2. In the Statistical analysis section, why the authors consider a p-value > 0.10 as a reference p-value to establish if there is an “obvious heterogeneity”? Why not 0.05?

3. In the Statistical analysis section, please substitute “methodological subgroup analysis” with “subgroup analysis” ;

4. In Study search section, please clarify “After detailed evaluation…”. Which kind of evaluation?

The authors mentioned the Newcastle-Ottawa Scale standard and evaluated the studies as “high-quality studies” with ≥7 points, but then they considered also studies with lower points. Why?

Moreover, there is only one recent study (i.e. from 2018) among the selected 15 studies;

5. In Table 1, the reference of the 15th study should be S.H. [31];

6. In Figure 1, there is a duplicated form at the end of the flow chart (n=15) and the first form “additional records identified through other sources” can be deleted (n=0);

7. In the caption of Figure 3, point D) is PR, not ER;

8. In the caption of Figure 4, the number of studies included is 15, not 11;

9. In the caption of Figure 5, the number of studies included is 15, not 11;

10. In Discussion, “D-dimer level has a certain auxiliary value…” should be substituted by “D-dimer level could have an auxiliary value…”.

Reviewer #2: In this paper the authors performed a systematic review and meta-analysis to evaluate the significance of D-dimer levels in the diagnosis and prognosis of breast carcinoma. The major limitation of this study lies on the difficulty to generalize the clinical value of D-dimer in breast cancer due to the fact that the authors considered mainly studies performed on Chinese population with only 2 studies with Caucasian subjects.

The paper is potentially interesting anyway a number of compulsory aspects need to be clarified and the following comments should be addressed before the manuscript can be considered for publication:

Major points:

• One critical aspect concerns inclusion and exclusion criteria which are poorly detailed: please clarify better what does “sufficient data” at line 88 mean and “no clear gold standard” at line 89. Moreover, please clarify in figure 1 the reason why the authors considered “not relevant” four papers in the step of eligibility, which is the criteria?

• In the paragraph “data extraction and quality assessment” the authors highlight the importance for a study to have NOS score ≥ 7. Why in this systematic review they consider also papers with NOS score of 5 or 6? Please justify this choice

• D-dimer levels are always missing unit of measure. Please specify.

• Some papers as for example Huang 2012, Xie 2011 and Kim 2004 showed D-dimer levels completely different from the other studies. Please discuss this aspect in the paragraph of “discussion”.

• In the Discussion section, at line 231 the authors said that plasma D-dimer levels can be used in prognosis of breast cancer. No data concerning prognostic significance are reported in the paper. Please modify and discuss better this part.

Minor points:

• Table 1: authors should include also details as regards samples used in different studies to quantify D-dimer levels. Always plasma with citrate as anticoagulant was considered?

• Figure 3A and 3E are the same figure, please modify.

Overall the manuscript can be interesting but it cannot be accepted in the present form. A major revision is mandatory before publication.

6. PLOS authors have the option to publish the peer review history of their article (what does this mean?). If published, this will include your full peer review and any attached files.

Reviewer #1: No

Reviewer #2: No

---

## [Author Response · Author response to Decision Letter 0]

24 Jul 2019

July 24, 2019

Dear Dr. Tagliabue:

We would like to thank you and the reviewers for your comments regarding our manuscript entitled “The association of D-dimer with clinicopathological features of breast cancer and its usefulness in differential diagnosis: a systematic review and meta-analysis” (ID: PONE-D-19-14829R1). 

The valuable insights have been helpful in revising and improving our manuscript, as well as in guiding the significance of our study. We have carefully reviewed the comments  and have accordingly revised the manuscript, which we hope meets your approval. The revised portions of the text are highlighted in yellow. The main corrections and responses to the reviewers’ comments are given below. We hope that the manuscript is now acceptable for submission to PLOS ONE. We look forward to hearing from you. 

Sincerely,

LY Zhang

Dongyang People’s Hospital

60 West Wuning Road, Dongyang 322100, Zhejiang, China

Tel: +8615267914600

Email: Happy_zhang1y@163.com

Responses to the reviewer’s comments:

Reviewer #1: 

1. Figure 3E is identical to Figure 3A. Please correct the duplicated figure;

We apologize for this error. The correct images for Figure 3 have been uploaded along with the revised manuscript.

2. As stated from the authors in the Discussion, the literature of this meta analysis is limited and a large number of patients is still needed. Moreover, they are also aware that there is the risk of a regional bias, since almost all studies are from China. Results are “somewhat restrictive”, as stated in the Conclusion.

We realize that this meta-analysis has limitations and have highlighted them in the discussion. We also searched for related articles for nearly 2months, but could not find new available research. However, we have included some references to the latest research on D-dimer and breast cancer. We will continue to track relevant information in this regard.

3. In the inclusion and exclusion criteria section, “the study results had a normal distribution” does not mean anything. Do the authors mean the distribution of the response variable?

Again, “the results were skewed” has no sense. Please explain it better;

Thank you for this comment. We apologize if the inclusion and exclusion criteria were ambiguous. Meta-analysis of continuous variables requires the extraction of the mean and standard deviation from each of the included documents. The median + quartile interval were generally provided because the results of the original study were skewed. According to the Cochrane Handbook, if the data are skewed, the median + quartile interval cannot be converted to mean and SD. However, because it did not make sense to emphasize the normal or skewed distribution, we only emphasized that the average and standard deviation could be correctly obtained. 

According to the principle of PICOS, the inclusion and exclusion criteria were modified as follows:

“The inclusion criteria were as follows: 1) the study group consisted of patients with breast cancer with a definite diagnosis; 2) the control group consisted of healthy women or patients with benign breast tumors; 3) the D-dimer test method in the study was clear; 4) the study results contained or had sufficient data to calculate the mean and standard deviation, defined here as more than 20 patients; and 5) the study showed a correlation between D-dimer levels and diagnostic and/or clinicopathological features of breast cancer. The exclusion criteria were as follows: 1) case reports or reviews; 2) studies describing animal experiments; 3) repeated publications; and 4) articles with a low Newcastle-Ottawa scale (NOS) score (≤4)."

4. In the Statistical analysis section, why the authors consider a p-value > 0.10 as a reference p-value to establish if there is an “obvious heterogeneity”? Why not 0.05?

According to the Cochrane Handbook, care must be taken when interpreting the chi-squared test results in meta-analyses because it has low power in (common) studies with small sample sizes or when only a few studies are included. This means that while a statistically significant result may indicate a problem with heterogeneity, a non-significant result must not be considered as evidence of no heterogeneity. Thus, we used the value of 0.10 instead of 0.05 to determine statistical significance.

5. In the Statistical analysis section, please substitute “methodological subgroup analysis” with “subgroup analysis” ;

Based on your comment, we revised the text as follows:

“In addition, when the heterogeneity was significant, we used a methodological subgroup analysis followed by a sensitivity analysis” was revised to “In addition, when the heterogeneity was significant, we performed a subgroup analysis followed by a sensitivity analysis.”

6. In Study search section, please clarify “After detailed evaluation…”. Which kind of evaluation?

According to the inclusion and exclusion criteria, only full-text articles were assessed for eligibility. We explained this in more detail in the revised manuscript text and also changed the phrase “After detailed evaluation...” to “After full-text articles were assessed for eligibility...”

The authors mentioned the Newcastle-Ottawa Scale standard and evaluated the studies as “high-quality studies” with ≥7 points, but then they considered also studies with lower points. Why?

We apologize for the ambiguity regarding the inclusion and exclusion criteria in the text. We excluded low-quality studies and added the following statement as an exclusion criterion: “Articles with low NOS score (≤4).”

According to the Cochrane Handbook, we used sensitivity analysis as an indicator of the document quality. The results of the sensitivity analysis were stable; therefore, we believe that these three studies can be included.

Moreover, there is only one recent study (i.e. from 2018) among the selected 15 studies;

Unfortunately, this was the only recent study that fulfilled the inclusion criteria. We will continue to track the latest relevant research. However, we hope that this meta-analysis will allow scholars to note the potential of D-dimer in breast cancer research.

7. In Table 1, the reference of the 15th study should be S.H. [31];

Thank you for this suggestion. We have modified it as recommended by the reviewer.

8. In Figure 1, there is a duplicated form at the end of the flow chart (n=15) and the first form “additional records identified through other sources” can be deleted (n=0);

We have modified Figure 1 as recommended by the reviewer.

9. In the caption of Figure 3, point D) is PR, not ER;

In the caption of Figure 3,(D) describes the graph of the relationship of D-dimer with the estrogen receptor (ER) status (positive vs. negative).Panel (A) reflects the relationship of D-dimer with the progesterone receptor (PR).

10. In the caption of Figure 4, the number of studies included is 15, not 11;

This meta-analysis includes the differences in the D-dimer levels between breast cancer patients and benign controls and healthy controls, as well as the correlation between D-dimer levels and clinical pathological characteristics of breast cancer. A total of 7 groups and 15 articles were included. The studies were used separately, and no study was used in every comparison. Although Biljana et al. mentioned in "Bias in meta-analysis and funnel plot asymmetry" that there is a limitation in the detection of bias with a small sample size, there is currently no standardization limit. We previously limited the publication bias test to “more than 10 included studies”; however, this may be ambiguous. 

Therefore, we deleted the original Figure 4/5 and published the bias for each group and presented it as the P value of the Egger test. The following figure is for reference for the Reviewer.

11. In the caption of Figure 5, the number of studies included is 15, not 11

Please see our response for Question 10. The following figure is for reference for the Reviewer.

12. In Discussion, “D-dimer level has a certain auxiliary value…” should be substituted by “D-dimer level could have an auxiliary value…”.

As per the reviewer’s suggestion, we have revised the corresponding statement.

We thank you for reviewing our manuscript.

Reviewer #2:

1.One critical aspect concerns inclusion and exclusion criteria which are poorly detailed: please clarify better what does “sufficient data” at line 88 mean and “no clear gold standard” at line 89.

Based on the reviewer’s suggestion, we included a definition of “sufficient data” in the revised manuscript as follows: “the study results contained or had sufficient data to calculate the mean and standard deviation, defined here as more than 20 patients.”

“No clear gold standard” implied that the study group included patients with breast cancer without a definite diagnosis. The exclusion criteria were originally based on the inclusion criteria, but we have now revised the exclusion criteria to avoid this ambiguity as follows:

“The exclusion criteria were as follows: 1) case reports or reviews; 2) studies describing animal experiments; 3) repeated publications; and 4) articles with a low Newcastle-Ottawa scale (NOS) score (≤4).”

Moreover, please clarify in figure 1 the reason why the authors considered “not relevant” four papers in the step of eligibility, which is the criteria?

Four papers were deemed to be “not relevant” after we browsed the full text because the studies assessed D-dimer and breast cancer but did not examine the aspects we wanted to study. For clarity, we have added the following statement as an inclusion criterion: “5) the study showed a correlation between D-dimer levels and diagnostic and/or clinicopathological features of breast cancer.”

After consideration, we believe that it can be combined with "data not available."

2. In the paragraph “data extraction and quality assessment” the authors highlight the importance for a study to have NOS score ≥ 7. Why in this systematic review they consider also papers with NOS score of 5 or 6? Please justify this choice

We apologize for the ambiguity regarding the inclusion and exclusion criteria in the text. We excluded low-quality studies and added the following statement as an exclusion criterion: “Articles with a low NOS score (≤4).”

According to the Cochrane Handbook, we used sensitivity analysis as an indicator of the document quality. The results of the sensitivity analysis were stable; therefore, we believe that these three medium-quality studies can be included.

3. D-dimer levels are always missing unit of measure. Please specify.

Currently, there is no uniform standard for D-dimer units, and the measurement units included in the study are different. Therefore, standardized mean difference was used as the effect combination index, and the effect of different measurement units on the results of the meta-analysis was negated.

We have described this in the discussion and have revised the D-dimer unit to ng/mL.

4. Some papers as for example Huang 2012, Xie 2011 and Kim 2004 showed D-dimer levels completely different from the other studies. Please discuss this aspect in the paragraph of “discussion”

Sensitivity analysis indicated that these three studies were not sources of heterogeneity. This complete difference may be because of the differences in detection methods and units. However, we agree with the reviewer’s suggestion and have discussed this aspect in the discussion as follows:

“At present, there is no uniform standard for the methods and units used to detect D-dimer levels, and the consistency between the results of the same test items in each laboratory is not strong. In this paper, the unified D-dimer unit was ng/mL, and the standardized mean difference was used as the effect combination index. However, inconsistent detection methods, reagents and type of anticoagulant may cause the absolute D-dimer value to differ greatly, leading to high heterogeneity among the literature results. Therefore, a uniform methodological standard should be established for D-dimer detection so that the data between different laboratories can be interoperable or comparable.”

5. In the Discussion section, at line 231 the authors said that plasma D-dimer levels can be used in prognosis of breast cancer. No data concerning prognostic significance are reported in the paper. Please modify and discuss better this part.

As the reviewer stated, because of the literature limitations, we did not have sufficient evidence to prove the relationship between D-dimer and breast cancer prognosis; therefore, we revised the text in the Background, abstract, discussion, and conclusion.

Background: “Studies have shown that D-dimer levels were significantly correlated with the diagnosis and prognosis of breast cancer.” was changed to “Studies have shown that D-dimer levels are significantly correlated with the differential diagnosis and clinicopathological features of breast cancer.”

Abstract: “The results of this meta-analysis indicate that plasma D-dimer levels can be used as an important reference for the early identification, staging, and prognosis of breast cancer” was changed to “The results of this meta-analysis indicate that plasma D-dimer levels can be used as an important reference for the early identification and staging of breast cancer.”

Discussion: “The plasma D-dimer levels can be used as an auxiliary index for the diagnosis, staging, and prognosis of breast cancer” was changed to “Elevated D-dimer levels suggest a worsening of the disease, a later clinical stage, and a greater likelihood of tumor metastasis. The plasma D-dimer levels can be used as an auxiliary index for the diagnosis and staging of breast cancer.”

“This evidence suggests that D-dimer has potential in the differential diagnosis and prognosis of breast cancer” was changed to “This evidence suggests that D-dimer has potential in the differential diagnosis and staging of breast cancer.”

We have also added the following text in the discussion: “Due to the limitations of the literature, the role of D-dimer in the clinical pathology and prediction of prognosis of breast cancer still needs to be studied in a large number of patients.”

6. Table 1: authors should include also details as regards samples used in different studies to quantify D-dimer levels. Always plasma with citrate as anticoagulant was considered?

According to there viewer’s suggestion, we have added the anticoagulant information in Table 1.Different anticoagulants also need our attention, but owing to the limitations of the included studies, subgroup analysis was not possible.

We discussed this in the discussion as follows:

“However, inconsistent detection methods, reagents and type of anticoagulant may cause the absolute D-dimer value to differ greatly, leading to high heterogeneity among the literature results. Therefore, a uniform methodological standard should be established for D-dimer detection so that the data between different laboratories can be interoperable or comparable.”

7. Figure 3A and 3E are the same figure, please modify.

We apologize for this error. The correct images for Figure 3 have been uploaded along with the revised manuscript.

We thank you for your comments on the manuscript.

---

## [Decision Letter · Decision Letter 1]

30 Jul 2019

PONE-D-19-14829R1

The association of D-dimer with clinicopathological features of breast cancer and its usefulness in differential diagnosis: a systematic review and meta-analysis

PLOS ONE

Dear Dr. Zhang,

Thank you for submitting your manuscript to PLOS ONE. After careful consideration, we feel that  the manuscript has been amended according to reviewers' concerns. However, as indicated by one of them, this meta analysis is limited and almost all considered studies are from China. Therefore, the authors should specify in the title and in the rest of the manuscript that the results refer to the Chinese population.

We invite you to re-submit a revised version of the manuscript that addresses this point.

We would appreciate receiving your revised manuscript by August 15, 2019. To enhance the reproducibility of your results, we recommend that if applicable you deposit your laboratory protocols in protocols.io, where a protocol can be assigned its own identifier (DOI) such that it can be cited independently in the future. For instructions see: http://journals.plos.org/plosone/s/submission-guidelines#loc-laboratory-protocols

We look forward to receiving your revised manuscript.

Kind regards,

Elda Tagliabue

Academic Editor

PLOS ONE

Reviewers' comments:

Reviewer's Responses to Questions

**Comments to the Author**

1. If the authors have adequately addressed your comments raised in a previous round of review and you feel that this manuscript is now acceptable for publication, you may indicate that here to bypass the “Comments to the Author” section, enter your conflict of interest statement in the “Confidential to Editor” section, and submit your "Accept" recommendation.

Reviewer #1: All comments have been addressed

Reviewer #2: All comments have been addressed

2. Is the manuscript technically sound, and do the data support the conclusions?

Reviewer #1: Partly

Reviewer #2: Yes

3. Has the statistical analysis been performed appropriately and rigorously? 

Reviewer #1: Yes

Reviewer #2: Yes

4. Have the authors made all data underlying the findings in their manuscript fully available?

Reviewer #1: Yes

Reviewer #2: Yes

5. Is the manuscript presented in an intelligible fashion and written in standard English?

Reviewer #1: Yes

Reviewer #2: Yes

6. Review Comments to the Author

Reviewer #1: All comments have been addressed.

The manuscript is well written and analyses have been performed following the proper methods for meta analyses.

However, I have some concerns since results are restrictive. The literature of this meta analysis is limited and a large number of patients is still needed. Maybe the authors should wait in order to collect more studies. MoreoverThe risk of regional bias should not be underevaluated, since almost all studies are from China (maybe the authors can specify in the title and in the rest of the manuscript that at the moment these results only refer to the Chinese population).

Reviewer #2: The authors have addressed the queries raised in my previous review.

The manuscript in the present form is suitable for publication.

7. PLOS authors have the option to publish the peer review history of their article (what does this mean?). If published, this will include your full peer review and any attached files.

Reviewer #1: Yes: Elena Landoni

Reviewer #2: No

---

## [Author Response · Author response to Decision Letter 1]

2 Aug 2019

August 2, 2019

Dear Dr. Tagliabue:

We would like to thank you and the reviewers for your comments regarding our manuscript titled “The association of D-dimer with clinicopathological features of breast cancer and its usefulness in differential diagnosis: a systematic review and meta-analysis” (ID: PONE-D-19-14829R1). 

Your comments have been helpful in revising and improving our manuscript. I have made the necessary minor revisions as per your suggestions. 

Of the 15 articles included, ten were from China and five were from countries outside China. Because our meta-analysis is based on published research, the fact that most of the data coming from China may have been perceived as a problem. Considering that the five documents from countries other than China are also main components of the article, they are distributed in each group of the study, so specification of the results referring to the Chinese population in the title may lead to increased limitations of the meta-analysis. 

We have adopted the reviewer’s opinion that "The risk of regional bias should not be under evaluated". We have added Figure 4, using subgroup analysis to evaluate the impact of region on heterogeneity, and discussed this issue in the “Discussion”. 

As follows, “The subgroup analysis was also used to examine the source of heterogeneity based on region with only significant differences in the benign control group. However, after excluding the study by Kim et al. of Korea from the benign control group, the heterogeneity between the eight Studies from China did not reduce, indicating that the regional differences cannot explain the heterogeneity between benign control groups."

We hope that this meta-analysis is not just another end point of research but that it encourages us to conduct more validation studies in other independent large groups, which will better clarify the association of D-dimer with breast cancer patients. This meta-analysis is limited, and we will continue to pay attention to the latest research in this area. When there is valuable literature, we will update it in time.

The minor revisions requested in the text are highlighted in yellow. The Figure 4 is given below. We hope that the manuscript is now acceptable for submission to PLOS ONE. We look forward to hearing from you. 

Sincerely,

LY Zhang

Dongyang People’s Hospital

60 West Wuning Road, Dongyang 322100, Zhejiang, China

Tel: +8615267914600

Email: Happy_zhang1y@163.com

Fig 4. Subgroup analysis of D-dimer levels and breast cancer-associated differential diagnosis and clinicopathological features according to region.Plots depicting comparisons between breast cancer patients and (A) benign controls, (B) healthy controls, (C) tumor node metastasis (TNM) stage (stage III-IV vs stage I-II), (D) lymph node status (positive vs. negative), and (E)estrogen receptor (ER) status (positive vs. negative). Group1: China; Group2: Regions outside China

---

## [Editor Report · Decision Letter 2]

6 Aug 2019

The association of D-dimer with clinicopathological features of breast cancer and its usefulness in differential diagnosis: a systematic review and meta-analysis

PONE-D-19-14829R2

Dear Dr. Zhang,

We are pleased to inform you that your manuscript has been satisfactory amended and judged scientifically suitable for publication. It will be formally accepted for publication once it complies with all outstanding technical requirements.

With kind regards,

Elda Tagliabue

Academic Editor

PLOS ONE
---

## [Editor Report · Acceptance letter]

27 Aug 2019

PONE-D-19-14829R2 

The association of D-dimer with clinicopathological features of breast cancer and its usefulness in differential diagnosis: a systematic review and meta-analysis 

Dear Dr. Zhang:

I am pleased to inform you that your manuscript has been deemed suitable for publication in PLOS ONE. Congratulations! Your manuscript is now with our production department. 

With kind regards,

on behalf of

Dr. Elda Tagliabue 

Academic Editor

PLOS ONE